# Epithelial to Mesenchymal Transition: A Challenging Playground for Translational Research. Current Models and Focus on *TWIST1* Relevance and Gastrointestinal Cancers

**DOI:** 10.3390/ijms222111469

**Published:** 2021-10-25

**Authors:** Luana Greco, Federica Rubbino, Alessandra Morelli, Federica Gaiani, Fabio Grizzi, Gian Luigi de’Angelis, Alberto Malesci, Luigi Laghi

**Affiliations:** 1Laboratory of Molecular Gastroenterology, IRCCS Humanitas Research Hospital, Via Manzoni 56, 20089 Rozzano, Italy; luana.greco@humanitasresearch.it (L.G.); federica.rubbino@humanitasresearch.it (F.R.); alessandra.morelli@humanitasresearch.it (A.M.); 2Department of Medicine and Surgery, University of Parma, 43126 Parma, Italy; federica.gaiani@unipr.it (F.G.); gianluigi.deangelis@unipr.it (G.L.d.); 3Gastroenterology and Endoscopy Unit, University-Hospital of Parma, Via Gramsci 14, 43126 Parma, Italy; 4Department of Immunology and Inflammation, IRCCS Humanitas Research Hospital, Via Manzoni 56, 20089 Rozzano, Italy; fabio.grizzi@humanitasresearch.it; 5Department of Biomedical Sciences, Humanitas University, Via Rita Levi Montalcini 4, 20072 Pieve Emanuele, Italy; alberto.malesci@hunimed.eu; 6IRCCS Humanitas Research Hospital, Via Manzoni 56, 20089 Rozzano, Italy

**Keywords:** EMT, plasticity, gastrointestinal cancer, stemness, tumor microenvironment, progression, translational research, chemo-resistance

## Abstract

Resembling the development of cancer by multistep carcinogenesis, the evolution towards metastasis involves several passages, from local invasion and intravasation, encompassing surviving anoikis into the circulation, landing at distant sites and therein establishing colonization, possibly followed by the outgrowth of macroscopic lesions. Within this cascade, epithelial to mesenchymal transition (EMT) works as a pleiotropic program enabling cancer cells to overcome local, systemic, and distant barriers against diffusion by replacing traits and functions of the epithelial signature with mesenchymal-like ones. Along the transition, a full-blown mesenchymal phenotype may not be accomplished. Rather, the plasticity of the program and its dependency on heterotopic signals implies a pendulum with oscillations towards its reversal, that is mesenchymal to epithelial transition. Cells in intermixed E⇔M states can also display stemness, enabling their replication together with the epithelial reversion next to successful distant colonization. If we aim to include the EMT among the hallmarks of cancer that could modify clinical practice, the gap between the results pursued in basic research by animal models and those achieved in translational research by surrogate biomarkers needs to be filled. We review the knowledge on EMT, derived from models and mechanistic studies as well as from translational studies, with an emphasis on gastrointestinal cancers (GI).

## 1. Epithelial to Mesenchymal Transition and Cancer in Pills

### 1.1. EMT and Multistep Carcinogenesis

The occurrence and progression of cancer, being a multi-factorial disease, implies multiple alterations of the prevailing processes involved in cellular and tissue homeostasis. Cancer cells present a mix of loss and gain of functions: normal traits are lost, and abnormal ones are acquired. Moving from gene changes (such as activating oncogenes and turning off tumor suppressors) and/or the way genes are transcribed and translated, the changes in genotype, transcriptome, and eventually protein function progressively lead to a malignant phenotype. The main structural and functional derangements that occur in cancer are summarized as its “hallmarks” [1,2]. However, clonal evolution does not imply metastasis formation and gene damage is not sufficient per se for metastatic competence, as demonstrated by several studies in which transgenic mouse models of cancer do not automatically establish distant metastases. Towards the ending of the golden era of molecular genetics, it become evident that genes acting alike “metastasis suppressor” had not been identified, nor they have been following the advent of next generation sequencing [3]. Transformed cells must therefore acquire additional abilities to overcome the barriers against their spread [4].

EMT was first conceived as a key process of embryogenesis and tissue repair in the early 1980s. In both processes, epithelial cells become activated by neighboring signals [5]. Accordingly, EMT participates in developmental and physiological processes and in an array of diseases and ensuing damage-response programs [6,7,8]. In cancer research, EMT hit the spot as a critical mechanism for the initiation of cancer cell transport, which may culminate in metastasis formation [9,10]. It could be somehow over-simplified as a process allowing a polarized epithelial cell, which interacts with basement membrane and maintains anchorage, to undergo biochemical and physical changes permitting the achievement of a mesenchymal phenotype. Such plastic transition encompasses the acquisition of migratory capacity together with invasiveness, and being a pleiotropic program, is also variably coupled with resistance to apoptosis, increased production of extra-cellular matrix (ECM) components [11], and possibly immune evasion and drug-resistance [12,13]. A wide range of molecules have been implicated in triggering the EMT program, and based on their actions they are classified as: transcription factors (TFs) which orchestrate the EMT programs (i.e., core regulators), extracellular signals which keep alive/activate this process (or inducers), and effector molecules which execute EMT programs (i.e., effectors) [14]. The phenotypical effect of EMT is the elongation of the epithelial cells, which become stretched and spindle-shaped, with fibroblast-like features, experience a loss of polarity, pseudopodia formation, and the breakdown of E-cadherin (*CDH1*) related cell-cell adhesion. In cancer cells, these changes would increase their mobility, favoring dissemination. Ideally, cancer cells leaving their primary sites need to adapt to a hostile environment and maneuver through their differentiation states by exploiting plasticity (Figure 1).

### 1.2. EMT and Tumor Microenvironment

The steps involved in the metastatic cascade are local invasion, intravasation, transport, extravasation, and colonization. As first step, tumor invasion is the process in which tumor cells move from the primary neoplastic nests to the nearby stroma [14].

Tumor progression comprises the detrimental contribution of host cells pushing towards the enhancement of cancer hallmarks [15,16]. Host cells surrounding the neoplastic cells belong to the peritumoral stroma, a complex entity referred to as “tumor microenvironment” (TME) [17,18]. It comprises, beyond tumor immune infiltrating cells, neo-vessels, extracellular matrix-associated molecules (such as cytokines and chemokines), and heterogeneous subtypes of cancer-associated fibroblasts (CAFs) which establish a pro-metastatic environment [9,19,20,21,22,23,24,25]. In interacting with the TME, transformed cells can modulate the functions of stromal cells, and vice versa, promoting their own growth and adaptation [26,27,28,29]. The TME is the source for a multiplicity of stimuli that promote and sustain EMT. Furthermore, the occurrence of EMT would impact on the perspective on TME, implying that it is not simplistically composed by “bad or good” host cells, but that it also may comprise tumor cells in mesenchymal disguise. If this is the case, pro-tumor effects ascribed to TME components might be reconsidered, as they could depend not solely on host cells surrounding cancer nests, but possibly also on tumor cells that look like stromal ones, or on the interaction of all these elements (Figure 2). Accordingly, it could be possible that some cells considered pro-tumoral (alike cancer associated fibroblast, CAF) might indeed be cancer cells in mesenchymal disguise, rather than reactive host cells. Despite the fact that this fascinating theory was postulated more than 20 years ago, only in recent years have human EMT cancer cells been revealed at the invasive front of solid tumors [30] and in the bloodstream of breast cancer [31], pancreatic (PDAC), and colorectal cancer (CRC) patients [32].

## 2. The Molecular Biology of EMT

Depending on the involvement in physiologic or pathogenetic processes, three types of EMT can be recapitulated. Type 1 is associated with implantation, embryo formation, and organ development and requires the reverse mesenchymal-to-epithelial transition (MET) to generate secondary epithelia. It has been speculated that such a layer may differentiate to form other epithelial tissues undergoing EMT and generating connective tissue cells [33]. The transition from epithelial cells to inflammation-induced fibroblasts can be observed in type 2 EMT, associated with wound healing, tissue repair, and organ fibrosis [34]. Eventually, type 3 occurs in neoplastic cells that have previously undergone genetic and epigenetic changes [35], specifically in genes that favor clonal outgrowth and the development of localized tumors. Type 3 EMT can be comprised in the process of local and distant progression, in which epithelial tumor cells variably acquire mesenchymal features while leaving their primary site to migrate [36,37]. As a complex process, the role of EMT in cancer has been partially elucidated by identifying its activators and thanks to cellular and animal models, mainly accomplished by turning on/off these activators themselves. As such, the models are likely extreme, as they exacerbate the phenotype of a complete transition which is seldom accomplished by cancer cells in humans.

### EMT and Crossing of Lineage-Specific Differentiation of Epithelial Cells in the Gastrointestinal Tract

The differentiation of intestinal cells is regulated at multiple spatial and temporal levels by an array of pathways, some of which participate in EMT. Along the gastrointestinal tract, Wnt and Notch convey signalling highly conserved among the species, from arthropods to vertebrates, which are implicated in the differentiation of secretory and absorptive cells and are often altered in EMT.

Wnt and Notch related signalling pathways have a high relevance in gastro-intestinal development and homeostasis, and are dysregulated in intestinal adenoma development (and chemoresistance against 5-fluorouracil and oxaliplatin in cancer cells) [38].

Wnt plays a pivotal role in embryogenesis and regulation of cell homeostasis by mediating the maintenance of intestinal stem cells and crypt compartments, by the downstream TF Achaete scute-like 2 (*ASCL2*) [39]. Signalling along a canonical Wnt pathway, triggered by the protein ligand, binds the relative Frizzled receptor, inhibiting the phosphorylation of β-catenin by the protein complex Axin, APC, GSK3, and CK1. This prevents the β-catenin degradation by ubiquitination and, in contrast, promotes its nuclear translocation and interaction with T cell factor (TCF)/lymphoid enhancer-binding factor (LEF) [40]. The pathway is almost invariably deranged by *APC* or *AXIN2* damage in CRC and is correlated with EMT in gastrointestinal tumors. High levels of *ASCL2* in gastric tumour downregulate *miR223*, consequently triggering EMT. This suggests the capacity of *ASCL2* to promote the invasion and migration of gastrointestinal cancer cells [41]. Other aberrations at different levels of the Wnt signalling could be critical for the CRC development and aggressiveness, due to an excessive activation of β-catenin [42].

Notch signalling also has a crucial role in the regulation of gastric and intestinal stem cell maintenance and epithelial cell differentiation. The pathway consists of the Notch receptors (Notch 1–4), their ligands (Delta-like (DLL) 1, 3, 4 and Jagged (JAG) 1, 2) plus the intracellular effector molecule NICD (Notch intracellular domain). NICD translocates to the nucleus where it recruits a transcriptional coactivator complex that activates the downstream transcription of Notch target genes, such as *Hes1* and *Olfm4* [43]. In the intestine, there are two different kinds of differentiated cell lineages: the absorptive ones, represented by the enterocytes, and secretory ones, comprising goblet, enteroendocrine, Paneth, and tuft cells. Here, the signalling activation has a determinant role in the cell programme differentiation. The activation of Notch signalling is implicated in the differentiation of absorptive enterocytes, while the secretory programme is promoted by its inactivation [44]. The family of Notch receptors comprises four different subtypes. Notch1 receptor and its ligand, Jagged1, have a critical role in triggering cancer cell stemness and invasive features. In particular, the abnormal activation of this pathway results in a high association with the activation on EMT in CRC, by the interaction of some relevant TFs, like *SLUG*, *SNAIL*, and their inducer *TGF-β* [45].

Thus, it appears that the same developmental pathways that are deranged along gastrointestinal carcinogenesis can connect neoplastic transformation to mesenchymal transition.

## 3. EMT Involvement in Gastrointestinal Cancers: The Frame

A pivotal role for the EMT has been demonstrated in the progression of CRC [46,47] and PDAC [48]. As reported in the Consensus Molecular Subtypes (CMSs) classification of CRC, based on comprehensive analyses of mRNA expression profiles, an EMT signature characterizes the CMS4 class [49] which is associated with stromal activation, immunosuppression, inflammation, angiogenesis, and is clinically marked by the worst outcome [50]. In CRC, the mesenchymal phenotype is strongly associated with migration, invasion, and metastasis [51]. The overexpression of fibronectin, a glycoprotein involved in cell interaction, and N-cadherin, a mesenchymal marker, were correlated with invasion [52] and metastasis [53], respectively, worsening patient survival. Fibronectin, particularly its alternative extra domain A (EDA), promotes angiogenesis and EMT combined with integrin α9β1 or by upregulating the autocrine secretion of VEGF-C in an PI3K/Akt-dependent pathway [54]. N-cadherin increases cell motility and migration by interacting with epidermal growth factor receptor (EGFR) 1 via the EC4 domain and by activating Ras-MAPK pathway and TCF/LEF transcription factor [55].

EMT is also induced by inflammatory cytokines and infiltrating macrophages, like TNF-α and IL-1β, TGF-β [56]. It has been shown that CRC cells with microsatellite instability (MSI) harboring frameshift mutations of the *TGFBRII* are less sensitive to TGF-β–induced EMT than MS-stable CRC cells, which have an intact TGF-β receptor type II [57]. Recently, the role of an inflammatory microenvironment created by inflammatory cytokines, growth factors, and macrophages was found to be crucial for inducing EMT and its reversibility in CRC cells [51], as well as in PDAC cells [58].

In addition, EMT can promote the development of chemoresistance and tumor recurrence [59,60] by different pathways [59,61,62,63,64], as well as the resistance to immune checkpoint blockade [65].

In a translational scenario, it would be advisable that improving our functional knowledge of the EMT-MET pathways could lead to the identification of molecular targets for innovative therapeutic interventions.

Recent studies focused their attention on the identification on EMT-based diagnostic and prognostic signatures in CRC [66,67] and PDAC [68]. Other have reported significant metabolic reprogramming events during EMT [69,70]. Metabolic rewiring can be overtly regulated by EMT-associated transcription factors (EMT-TFs) [71]. It is reasonable to consider that along various EM transition states neoplastic cells exploit distinct cellular metabolic systems. Accordingly, targeting EMT could be possible with the identification of vulnerabilities in metabolic pathways by the use of specific inhibitors, coupled with standard chemo- or immunotherapy [72].

## 4. EMT-Related Pathways

The entrance into the process of EMT is orchestrated by different TFs, mostly able to suppress *CDH1*. These players are members of the SNAIL zinc-finger family, including *SNAI1* (*SNAIL*) and *SNAI2* (*SLUG*), distantly related members of the family of zinc-finger E-box-binding homeobox *ZEB1* and *ZEB2*, and the basic helix–loop–helix (bHLH) family of TFs *TWIST1*, *TWIST2*, and *E12/E47*. Once the mechanisms controlled by *CDH1* are inhibited, epithelial cell characteristics are restrained and the switch towards a mesenchymal state promoted. The EMT process is further regulated by signaling pathways in which transforming growth factor (TGF)-β, Wnt, Notch, and receptor tyrosine kinases (RTKs) (co-)operate to modulate EMT response. TGF-β induces phosphorylation and activation of SMAD1 and SMAD2 nuclear proteins to trans-activate *SNAI1* expression. Otherwise, SMAD4 is an important negative regulator in this pathway, suppressing signal transducer and activator of transcription 3 (STAT3), which may directly contribute to the EMT process and to the overexpression of other TFs, alike *ZEB1* in CRC progression [73]. *ZEB1* and *SNAI1* are also directly activated by transcription factor 4 (*TCF4*) along Wnt signaling, promoting EMT through the inhibition of glycogen synthase kinase-3β (*GSK3β*) to stabilize β-Catenin, which translocates to the nucleus and therein actives transcription [73,74]. However, Notch and Nuclear Factor κB (NF-κB) signaling disrupt the GSK3β-SNAI1 interaction [75,76]. On the other side, Axin2, a canonical Wnt suppressor, acts as a potent tumor promoter by up-regulating the activity of *SNAI1*. Other inducers of the EMT comprise epidermal growth factor (EGF), fibroblast growth factor (FGF), and hepatocyte growth factor (HGF) which act through receptor tyrosine kinases (RTKs). These growth factors activate the major RAS-RAF-MEK1/2-ERK1/2 MAPK signaling cascade and promote *SNAI1* and *SLUG* expression [77]. The intracellular domain of Notch is even able to activate *SNAI2* expression [78], and its inhibition partially reverts the EMT process, reducing invasiveness [79]. These mechanisms are summarized in Figure 3.

Furthermore, small non-coding microRNAs (miRNAs) affect the expression of up 30% of genes [80], contribute to pathway regulation and also affect EMT [81]. In fact, miRNAs can act as tumor suppressors as well as oncogene depending on their target [81,82] and also have an important role in the regulation of plasticity of cancer cells. Particularly, a crucial role for the members of the miR-200 family has been suggested [83,84]. Amongst the prominent target genes of miR-200 are *ZEB1* and *2* [84,85], direct repressors of the epithelial marker *CDH1*. In patients with PDAC, high levels of miR200a and miR200b were detected in the serum [86]. In addition, the downregulation of miR-141 was observed in gastric cancer tissue compared to normal counterparts [87]. Then, a recent study suggests that the increased expression of miR-200c results in the negative regulation of its target genes including *ZEB1*, which as consequence modulates *CDH1* and *VIM* expression to promote EMT in CRC cells [82]. Not only ZEB, but also the Snail family is differentially regulated by miRNAs. Interestingly, the members of miR-200 family, the miR-182 cluster, miRs-18a, -18b, -31, -99, -100, -133, -203, -223, -375, and -486 were all significantly downregulated in the epithelial Madin-Darby canine kidney (MDCK) cells [88,89]. Several miRNAs are as well involved in the regulation of TWIST proteins. miR-10b expression in breast cancer cells can activate the hyaluronic acid (HA)/CD44 axis in TWIST-dependent way with the descending downregulation of the tumor suppressor gene *HOXD10*, a critical prerequisite step for the acquisition of metastatic properties [90]. In dissecting miRNAs that were differentially expressed in gastric cancer, Li and colleagues identified miR-223 as overexpressed in metastatic gastric cancer cells only, while it stimulated migration and invasion in non-metastatic ones. They found that miR-223 was induced by *TWIST* via binding to an E-box located in its core promoter, then binding to the 3′UTR of erythrocyte membrane protein band 4.1-like 3 (*EPB41L3*) and suppressing its translation [91].

EMT can be even induced by growth factors (such as TGFβ) and cytokines such as IL-6, which can be secreted by tumor cells but also activated by tumor-associated macrophages (TAMs) [92,93]. The role of TAMs, either blocking or enhancing tumor formation and/or progression is likely organ dependent, and their impact on EMT is still unclear. However, it is generally accepted that TAMs infiltration in gastrointestinal tumors is usually associated with an aggressive phenotype, and advanced clinical stage [94,95], although this is likely not the case in CRC [21,96].

### TWIST1, a Molecular Culprit

Among the main EMT TFs, such as *SNAI1*, *SNAI2*, *ZEB1* and *ZEB2*, *TWIST1* acts as a master regulator of several EMT-associated processes [97]. *TWIST1*, a highly conserved basic helix-loop helix transcription factor, plays a fundamental role in embryonic development, tissue differentiation and organogenesis. In parallel, its abnormal expression resulted involved in EMT and in cancer progression. Among interacting pathways, *p53* has a major relevance.

Oncogenic insults inducing *p53* and/or retinoblastoma (*Rb*) expression would trigger apoptosis or senescence, as barriers against cell transformation and tumor initiation [2,98]. *TWIST1* overexpression suppresses cellular senescence in response to genotoxic damage and sustains the proliferation of cells with accumulation of DNA damage [99]. Vichalkovski and coll. demonstrated that *TWIST1* phosphorylation at *Ser42* by *PKB/AKT2* inhibits *p53* in response to DNA damage, and such post-translational modification enables *TWIST1* activation during carcinogenesis [100]. Alike, Maestro’s group discovered that *TWITS1* can reduce the expression of the *ARF* tumor suppressor, indirectly affecting *p53* through the *ARF/MDM2/p53* axis [101]. Consistently, Valsesia reported that the over-expression of *TWIST1* is responsible for ARF/p53 inhibition in Myc-dependent apoptosis in neuroblastomas [102]. Piccinin et al. also demonstrated that *TWIST1* can directly bind *p53* C-terminus, facilitating *MDM2*-mediated degradation [103]. Along the line enhancing the effects of erased tumor suppressor guardians in cancer cells, *TWIST1* can also interact with *HOXA-5*, negatively regulating *p53* gene expression at transcriptional level [104]. Similarly, Pinho discovered that *p53* -/- pancreatic epithelial cells undergo EMT and express high levels of various TFs, comprising *Snai1*, *Snai2*, *Twist*, *Zeb1,* and *Zeb2* [105]. The study raised the possibility of a mutual regulation between *TWIST1* and *p53*, filling the gap between *p53* inactivation and *TWIST1*-induced metastasis. This was even supported by the study of Ansieau’s group, demonstrating that *TWIST1* overrides oncogene-induced senescence while inducing the EMT [106]. However, Beck et al. also found that low levels of TWIST1 may control tumor initiation in both a *p53*-dependent and -independent manner without inducing EMT, suggesting that *TWIST1*-induced tumor initiation and EMT are not necessarily functionally related [107]. *TWIST1* can enhance tumor initiation, conferring resistance to senescence and apoptosis, promoted by *p53* [108]. In addition, the combination of Notch pathway activation with the loss of *p53* function triggered the expression of EMT-TFs, including *Twist1*, in mouse gut, hence promoting an invasive and diffusive behavior [109].

Over the ability to promote EMT, cancer progression and invasion, TWIST1 also has a relevant role in the reduction of the sensitivity to chemotherapy in some CRC cell lines [110]. This TF can induce multi-drug resistance by upregulating *ABCB1* and *ABCC1*, members of the superfamily of ATP-binding cassette (*ABC*) transporters, which act as drug efflux pumps [111]. Further, high levels of TWIST1 in resected human CRC specimens were associated with the expression of P-gp, a transmembrane glycoprotein conferring multidrug-resistance phenotype to cancer cells [110]. Altogether, these data suggest the possibility to assess whether *TWIST1* is a potential target for re-establishing an apoptotic response and enhance the sensitivity to chemotherapy in CRC.

The high expression levels observed in CRC support TWIST1 as a prognostic biomarker in this tumor. However, the correlation of TWIST1 expression levels with lymph node metastasis and stages remains controversial, since some studies highlighted a significant relationship, while others did not. It is interesting to note that TWIST1 was reported to be a poor prognosis factor some years ago, being associated with the poorest overall and disease-free survival in stage I–II CRCs. A possible explanation for the loss of prognostic features in more advanced stages can be the accumulation and overlap of other deranged pathways to TWIST1 overexpression [112]. Alternatively, one can simply reason that a prognostic feature in early-stage CRCs is highly meaningful on the clinical ground, as it identifies the cases with the highest metastatic potential before it actually takes place, while molecular differences are later smoldered by the clinical behavior of the disease and the superimposed treatments. Recently, some of these inconsistencies have been partially explained by noticing that many works focused on the cytoplasmatic rather than the nuclear expression of TWIST1, which as TF acts in the nucleus. Consistently, high nuclear expression levels of TWIST1 resulted in being significantly associated to a shorter disease-specific survival and progression-free survival, suggesting the relevance of the TWIST1 subcellular localization for its effective prognostication [113].

Searching for epigenetic alterations is a relevant option for the identification of biomarkers [114,115]. As such, DNA methylation is frequently examined in various cancer, including CRC [116]. In 2010, a Japanese study reported increased *TWIST1* methylation levels in CRCs and adenomas vs. normal mucosa (median 55.7%, 25.6%, and 0.0%, respectively), and methylated *TWIST1* was suggested to be a potential biomarker in early CRC, with 89.6% of accuracy [117]. In parallel, higher expression levels of *TWIST1* mRNA were associated with a worse outcome, although, rather surprisingly, no association was detected between *TWIST1* methylation and its expression. On the other side, Lin et al. examined 353 plasma samples from CRC patients, finding that 70% had *TWIST1* hypermethylation without significant prognostic implication (with hazard ratios of 1.06 and 0.79 respectively, for univariate and multivariate analyses of disease-free survival) [118]. These contradictory findings cast doubts on *TWIST1* methylation as a reliable prognostic marker. Another work evaluated *TWIST1* (and *TWIST2*) methylation status in six well-established colorectal cancer cell lines (HTC116, COLO205, HT29, SW620, HCT15, LS180), detecting hypermethylation in all lines (range *TWIST1*: 52.3–94.1%). This study suggests that the pro-tumor effects of *TWIST1* might mainly come from the tumor stroma [119]. These data suggest that TWIST1 expressed in tumor stroma could contribute to EMT-like tumor budding phenotype, and partially support a possible role for its promoter methylation as a regulatory mechanism in CRC.

## 5. Models to Unravel the Role of EMT in Disease Progression

Despite the growing evidence that EMT takes place in human cancer, several important points remain to be addressed. First, it would be crucial to understand not only how EMT starts, but also to better clarify the molecular hierarchy and pathways that regulate it. This is difficult to unravel due to the fact that the main players of EMT are non-redundant and tissue specific, so that hierarchies and roles may change depending upon the tissue of origin of the primary tumor and the distant site of seeding [120]. Furthermore, the pleiotropic nature of EMT may also confer stemness properties to the cells entering in a plastic state, alongside with immune-evasive features. Overall, CSC features fit the capability of cancer cells to travel far from the site of origin, surviving anoikis, escaping immune surveillance, to land in a niche and eventually, for a very minor fraction of them, to grow into mass forming secondary lesions. Coherently, once we accept the notion of EMT, how it turns into its reversal, that is MET, is the descending issue. With respect to the original view, in which both EMT and MET should be fully accomplished, the current view is flexible, and intermixed E-M states tightly related to stemness are currently envisioned.

### 5.1. In Vitro Modeling of EMT

A critical issue is the paucity of cancer cell-lines from epithelial tumors displaying mesenchymal features. The availability of similar cell-lines would support EMT, further allowing functional studies employing cells with an array of mesenchymal features [30]. In humans, few primary cancer cells with such features have been isolated so far from epithelial cancers. Accordingly, most studies on EMT features of human cancer cells have been performed by genetic engineering, or by exogenous stimulation with EMT inducers, chiefly TGFβ and hypoxia. The favoring effects of hypoxia on EMT can be reversed by inhibiting HIF-1α receptor. These data indicate that either metabolic pathways or related relay molecules are involved in the interconversion process [121].

In hunting for epithelial cancer cells, one would discharge anything else looking like a fibroblast, which would be considered as a contaminating element in the party. The recovery of these cells may require a sort of radically different subtractive approach, in which epithelial cancer cells are discharged and residual mesenchymal cells subsequently tested for their neoplastic origin. In any case, some human cancer cells with variably pronounced mesenchymal features have been described, e.g., COLO741 and SW620 CRC cell-lines [30]. More recently, a CRC cell-line was isolated from patient blood, moving from CTCs, and named CTC-MCC-41 [122]. Very interestingly, this cell-line displays an intermediate E-M phenotype together with stem-cell like properties, making it a proof of principle that EMT occurs in invasive CRC cells able to invade the bloodstream and that at least a fraction of these cells also has stemness attributes.

In vitro methods to study EMT include cell characterization, functional assays, and EMT induction. A key aspect to functionally characterize EMT is related to wound healing and the migration of mesenchymal-like cells at the injury site. Standardized assays, e.g., cell invasion and scratch assays [123], offer an effective way to monitor the migration/invasion of mesenchymal-like cells, assessing the efficiency of EMT (induced or not), as well as investigating the mechanisms underlying re-epithelialization [124].

A common model to study EMT in vitro includes monolayer cultures of cancers cells in CAFs medium [125]. Other models employ cancer cells co-cultured with ECM proteins or with stromal cells [126], trying to unravel the involvement of these interactions in the induction of EMT. Although 2D studies represent an easy and moldable way to analyse EMT-related effects, they fail in reconstructing the complex environment in which the process takes place. One of the major criticisms concerning studies in 2D is that when cells are removed from their biological environment, they show an un-natural behaviour [127,128]. In contrast to these conventional systems, more advanced models have been developed trying to mimic the in vivo architecture of tumors, by mixing tumor 3D spheroids with fibroblasts [129]. Mina Bissell [130] was a pioneer in using 3D culture systems to model the molecular mechanisms underlying breast cancer cell invasion, revealing that epithelial tumor cells cultured in Matrigel change shape, resembling what happens in in vivo tumor progression. To capture the phenotypic switch of cells during EMT, a model has been established in which TGFβ-1 supply can efficiently generate mesenchymal-like cells, while its removal resulted in the re-acquisition of an epithelial-like phenotype [131,132].

Since then, several systems have been developed to mimic the in vivo microenvironment in an in vitro context, such as the use of permeable filter supports, 3D organotypic matrices (collagen, Matrigel), and co-culture systems in spheroids and organoids [133,134]. When embedded within organotypic matrices, most epithelial cells, such as those in mammary tumors, are able to recapitulate important features like formation of growth arrested ducts and acini with hollow lumen as well as collagen IV and laminin V basal deposition [135].

Although the composition of the stem cell niche varies from organ to organ, the organoids culture of the human GI epithelial tissue commonly requires Wnt activators (Wnt 3A and R-spondin), EGF and FGF, a BPM inhibitor (Nogging) and TGF- β inhibitor, together with additional tissue-specific factors [136,137]. Therefore, the organoid systems may show limitations to study biological processes requiring physical interaction between differentiated and non-differentiated epithelial cells [138]. This drawback can be partially overcome by modifying the culture method, e.g., by using different growth factors [139], by co-culturing organoids with stromal [140,141,142] or immune cells [143,144], or using of air–liquid interface culture [145,146].

A study that compared seven PDAC cell lines grown as 2D monolayer as well as 3D-spheroids showed strong CDH1 and β-catenin immune-reactivity at cell–cell boundaries, suggesting that adherent junctions are still present in both experimental conditions. The epithelial differentiated phenotype was strongly confirmed, especially in 3D model [147]. Finally, it is currently recognized that adding the 3rd dimension creates significant differences in cellular characteristics and function [148], demonstrating the relevance of 3D systems to investigate cancer molecular mechanisms including EMT.

### 5.2. In Vivo Animal Models of EMT⇔MET, and Stemness Features

To clarify the contribution of EMT to the mechanisms of cancer progression, we will discuss the controversial evidence provided by genetically engineered animal models. These models are based on knocking out/in key regulators of EMT (i.e., EMT-TF genes) or exploiting EMT lineage tracing models (Figure 4).

Two models targeted *Snai1* and *Twist1* to control EMT in breast and squamous carcinoma models, respectively [149,151]. In the first study, the results highlighted that endogenous *Snai1* expression was sufficient for breast cancer metastasis, also showing that *Snai1* expression in breast cancer was transient and able to block MET process [151]. In line with these findings, Tsai et al. showed that *Twist1* upregulation in a squamous cell carcinoma mouse model promoted invasiveness, and its silencing allowed MET. All these data are consistent with results from previous in vitro studies [149,154].

The above-mentioned studies support the role of EMT/MET in metastasis in vivo, while other data point to a role of EMT in chemoresistance [152,154], as further open-question.

Using genetic mouse model mimicking the course of human PDAC, Zheng et al. [152] reported that genetic suppression of *Snail* or *Twist1* in PDAC did not affect neither tumor progression, neither metastasis formation. This could be warranted by a compensation mechanism carried out by other EMT-TFs, as described in in vitro models [88]. Another hypothesis is that EMT-TFs expression is organ (and therefore) tumor specific, e.g., *Snai1* and *Twist1* are not necessary in mouse model of pancreatic cancer, while *Zeb1* knock-out reduces metastasis of about 30% [153]. Interestingly, this study also showed that *Zeb1* deletion causes a suspension of invasion and metastasis of pancreatic tumor cells.

Another in vivo model of breast cancer, based on Twist1 overexpression in the context of H-Ras activation, led to a highly undifferentiated tumor with EMT features [155] and claudin-low phenotype [150].

One elegant and unsurpassed PDAC model used the yellow fluorescent protein (YFP) to label and follow invading tumor cells [156]. Noticeably, CTCs were detectable in the bloodstream in animals with precancerous pancreatic intraepithelial neoplasia (i.e., PanIN) lesions, showing a switch from E-cadherin to N-cadherin expression coupled with *Zeb1* overexpression. Interestingly, this study supports that EMT occurs early in pancreatic neoplastic lesions. In fact, YFP+/Cdh1– cells have tumor-initiating capacity and are able to generate both Cdh1 positive and negative cells [156].

A similar approach, was used in a mouse model for intestinal tumors with conditional activation of Notch receptor (labelled with green fluorescent protein (GFP)) and *p53* inactivation [109], mimicking aggressive CRC. The expression of several EMT-TFs were detected in invasive and desmoplastic regions, while Zeb1 expression was detected in invasive GFP+ cells undergoing EMT [47]. In vivo and ex vivo analysis allowed the identification of GFP+ EMT-like cells at metastatic sites supporting the validity of this genetic model to study epithelial plasticity, as well as the association of *Notch* activation (in the context of *p53* downregulation) with metastatic CRC [109].

By inducing clonal PDACs by CRISPR/Cas9 somatic mutagenesis, it is notable that the mesenchymal phenotype was associated with *Kras^G12D^* expression, whose overexpression in hPDAC cell lines induced an EMT signature, with Vimentin upregulation and Cdh1 repression [157].

Models accomplished by genetic manipulation and/or by EMT inducers were pointing to achieve a complete transition. However, this evidence should be reconsidered, with plasticity being part of the process together with its pleiotropic features, including the acquisition of stemness like capabilities of cancer initiating cell (CIC, or cancer stem cells, CSC).

Since the importance of organotropism has been recently highlighted, a paucity of studies addresses the topic. Based on “Seed and Soil” theory [158], cells evolve in primary tumors acquiring metabolic signature supporting their future colonization in a particular organ. It has been proposed that only “metabolically flexible” cancer cells can survive in distant organs and form metastases [159]. A recent study identified a subset of CSCs with specific metabolic signature in human and mouse pancreatic tumors, demonstrating how they preferentially colonize and survive in distinct organs (i.e., liver or lung) [160]. Lung metastases showed ALDH+/CD133+ CSC phenotype, oxidative metabolism, MET-like phenotype, and were CD44-low cells. On their side, liver metastases showed drug-resistant phenotype, glyco-oxydative metabolism, EMT-like phenotype, and were CD44-high cells [160].

Reichert et al. provided a new perspective through which to view metastatic organotropism in PDAC. They demonstrated that, in a *Kras^G12D^* background, the absence of p120catenine (*p120ctn*) mediated epithelial identity and had variable adverse consequences on the localization of metastases. While mono-allelic loss of *p120ctn* accelerates *Kras^G12D^*-driven PDAC formation and liver metastasis, biallelic *p120ctn* loss was permissive for lung metastases [157,161].

Recently, Zhang and colleagues presented evidence that hypoxia induces the expression of lncRNA CASC9 and cascade activation of glycolysis and activation of the EMT process in various PDAC cell lines. Moreover, knockdown of CASC9 inhibited the tumorigenicity and metastasis in vivo, providing a potential target for pancreatic cancer treatment [162].

## 6. Tissue Expression of EMT-TFs and Their Potential as Biomarkers of Disease Progression in Translational Studies

Several markers have been put forward for the tissue fingerprinting of EMT, although it should be remarked that it cannot be interpreted like a black and white portrait. In fact, the plasticity of EMT implies that most cells entering into the program would have a dual, mixed epithelial-mesenchymal expression profile [163]. An array of studies evaluated the expression of selected EMT markers by immunohistochemistry to address their changes during tumor progression.

Most translational studies showed that the upregulation of surrogate EMT markers correlates with a worse prognosis. However, a few markers are currently considered not appropriate to assess EMT status, as functional changes should also be addressed [164]. While this is view is functionally appropriate, it clearly poses limitations to translational studies assessing EMT in human specimens.

### 6.1. Immunohistochemical Assessment

The critical issue of immuno-histochemical studies and the correlative associations with patient survival or with tumor features is that these provided no proof that the expression of such factors in the stromal compartment is to be ascribed to neoplastic cells, rather than to host cells of the tumor microenvironment. In any event, cells expressing EMT-TFs in the TME could be a mixture of stromal and cancer cells, and their ratio is difficult to establish. The differentiation of cancer cells dressed up with mesenchymal features from reactive host cells represents an experimental issue with intrinsic difficulties. Hybridization experiments show that only a fraction of stromal cells expressing TWIST1 also shares the same genetic alterations of cancer cells. Second, EMT cancer cells may express a variety of TFs in a non-concerted way [30].

It is noteworthy that stromal fibroblasts harboring the same chromosomal alterations of cancer cells were serendipitously described two decades ago in CRC. Such alterations, e.g., chromosome 7 trisomy, were then been interpreted as a cancer “field-effect”. Based upon the evidence that a fraction of TWIST1+ stromal cells harbor the same chromosomal changes of epithelial cancer cells, we could now re-interpret those findings and trisomic fibroblasts as being indeed EMT cancer cells in the peritumoral stroma. Interestingly, a correlation between the occurrence of chromosome 7 trisomy and the presence of TWIST1+ cells in the stroma/cancer has been reported also in breast cancer [165].

Correlative evidence between the expression of EMT-TFs and tumor molecular features comes from CRC. It has been shown that CRC with defects of the DNA mismatch repair system and hence microsatellite instability (MSI) do not express EMT-TFs and are unresponsive to the EMT-inducer TGF-β, due to frameshift mutations of the TGFβRII gene, which abrogate the expression of the receptor in MSI cancer cells [57]. In human CRCs, the expression of EMT markers was significantly associated with adverse clinicopathologic features and MS-stable phenotype. These findings define a genotype–phenotype relationship between *TGFbRII* and EMT which may contribute to the better prognosis of MSI CRCs [166,167]. This scheme well fits with the lower metastatic potential of MSI CRC [168], and their low rate of post-surgical progression with respect to MSS CRCs [169]. This behavior is also in line with the less frequent expression of TWIST1 in the peritumoral stroma of MSI CRC as compared to the MSS ones [30]. Thus, one should ask whether EMT is more pronounced in cancers with a low mutational burden than in those with high one. However, a positive answer is only partially supported by data [170], and this field deserves further investigation. Considering that the link between adaptive immune response and mutational burden is well established, it is reasonable to hypothesize that cancers with low mutational burden, and thus lower immune response and high progression rates [166,167], would also be the ones with more pronounced EM plasticity.

Although *APC*, *KRAS*, *TP53,* and *PIK3CA* were frequently mutated, most synchronous colorectal cancers (syCRCs) show different mutation profiles compared with single colorectal cancer, and have a worse outcome if MS-stable [171,172]. Wang et al. performed whole-exome sequencing to characterize the genetic alterations in syCRCs. Lesions from same patient showed distinct landscapes of somatic aberrations and shared few mutations, which suggests independent origin and development, despite the similar genetic background. They identified a recurrent somatic alteration (*K15fs*) in RPL22 in 25% of the syCRCs. By functional analysis, it appeared that mutated *RPL22* can suppress cell apoptosis and promote EMT [173].

Other studies addressed cytokeratin, mucin-2, and trefoil factor 3 as epithelial markers, while cystic fibrosis transmembrane conductance regulator, zinc finger E-box-binding homeobox 1 or 5-hydroxytryptamine receptor 2B staining were evaluated for distinguishing the mesenchymal phenotype [49,174].

*ETS1* (v-ets erythroblastosis virus E26 oncogene homologue 1) is a TF regulating RhoC expression during EMT and, consequently, CDH1 expression or its delocalization from adherent junctions which may act as a biomarker of adverse outcome in CRC [175]. Bates et al. identified tumorigenic properties for αVβ6 integrin, a receptor for fibronectin. In early-stage CRC, the activation of autocrine TGF-β is promoted by αVβ6 integrin, sustaining EMT, and is required for the migration guided by interstitial fibronectin [176].

In addition, the tumor suppressor gene *SMAD4* participates in EMT, as it is known to complex with multiple TFs (e.g., *SNAIL*, *SLUG*, *TWIST1*), in various types of cancer, promoting the repression or activation of target genes [177].

TWIST1 protein contains two functional nuclear localization subunits, suggesting that it is likely to be actively anchored in the cytoplasm on compliant matrices, therefore preventing nuclear translocation mediated by β1 integrin activation [178,179,180,181,182,183]. The association between TWIST1 nucleocytoplasmic expression with the clinicopathologic parameters and survival outcomes was evaluated by a tissue microarray (TMA)-based immunohistochemistry technique [184].

Galván and colleagues first showed that TWIST1 and TWIST2 protein expression are found nearly exclusively in the tumor stroma. They have shown that only in the zones of tumor budding high-expression of TWIST1 and TWIST2 had the pattern of the classic EMT hallmarks, including the nuclear accumulation of β-catenin, disrupted CDH1, and over-expression of ZEB1 and ZEB2 in stromal cells. Besides, TWIST1 overexpression in CRC cells was associated with more advanced pT, lymph node metastasis, and worse survival [119].

In consideration of these evaluations and to clarify the prognostic value of the histological category of EMT in colorectal cancer (CRC), Ueno et al. [185] proposed a model to categorize EMT by the integrated assessment of dedifferentiation and desmoplastic environment as a potent prognostic index independently of staging factors.

### 6.2. Expression Studies of mRNA Profiles

Sun et al. [186] evaluated the expression levels of the *TUG1* long non coding (lnc)RNA whose overexpression appears to be related to the silencing of the histone deacethylase 1 gene (*HDAC1*) in cell lines and clinical specimens of primary CRC. *TUG1* overexpression promoted the aggressiveness of CRC, increasing their colony formation, migration, and invasion in vitro, enhancing their metastatic potential in vivo, the opposite being accomplished by its knockdown. High *TUG1* levels in CRC tissues were significantly associated with a shorter survival in CRC patients, suggesting that *TUG1* might be an independent risk factor for CRC metastasis.

Poor prognosis is also related with overexpression of members of the SNAIL zinc-finger family of TFs. In 2006, Shioiri et al. [187] proved that high levels of *SLUG* mRNA act as crucial regulators of EMT by suppressing epithelial markers and adhesion molecules including *CDH1* in CRC. The expression of *SLUG* was significantly associated with TNM stage and distant metastasis and had a significant impact on patient overall survival. *SNAIL* is also necessary for the pro-tumorigenic effects of fibroblasts on colon cancer cells. In co-xenografted nude mice, the expression of *SNAI1* in stromal fibroblasts was required for the pro-tumorigenic effect of these cells on colon cancer growth and invasion. Furthermore, a direct association between stromal *SNAI1* expression and that of the endothelial marker CD34 was observed in tumor specimens from CRC patients [188]. Later, gene set variation analyses indicated that *SNAIL* expression strongly affects features related to cell cycle and Wnt/β-Catenin signaling. This correlated with the upregulation of the lymphoid enhancer-binding factor 1 (LEF1), a nuclear binding partner of β-Catenin. In the same way, *SNAI1* expression positively correlates with *LEF1* expression in CRC tissue and cell lines, and high levels of *SNAI1* have been found to be associated with patient mortality [189].

As to CRC, Guinney et al. [49] in their “Consensus Molecular Subtypes” defines 4 subclasses of tumor, including EMT colorectal tumor in a class called “CMS4”. This subclass includes tumors in which EMT genes are upregulated together with other genes implicated in activation of TGF-β signaling, angiogenesis, matrix remodeling pathways, and the complement-mediated inflammation. In addition, CMS4 specimens exhibited a gene expression profile compatible with stromal infiltration. Similarly, Roepman et al. [190] recapitulated CRC into three types. A-type CRC showed a significantly reduced expression of two mesenchymal markers (i.e., TWIST1 and AXL Receptor Tyrosine Kinase), but also a reduction in one epithelial marker (i.e., CDH1). In B-type cancers, four of the five epithelial markers were up-regulated and three mesenchymal markers (CDH2, Fibroblast Growth Factor Receptor 1 (FGFR1), and TGFβ1) were down-regulated. In C-type cancers, almost all the mesenchymal markers, except FLT1, were significantly up-regulated, and three epithelial markers (CDH1, Epidermal Growth Factor Receptor (*EGFR*), and *MET* Proto-Oncogene, Receptor Tyrosine Kinase) were down-regulated. These data indicate that type A and B correlate with epithelial phenotype, while type C cancers are mostly mesenchymal [191].

A recent study by microarray analysis on a mRNA dataset derived from GSE81582 based on the GPL15207 platform (Affymetrix Human Gene Expression Array) suggests that differentially expressed genes (DEGs) and miRNAs (DEMs) may provide the basis for further understanding the mechanism of CRC metastasis. Cai et al. demonstrated by survival analysis that low-expressed growth arrest specific 1 (GAS1)-associated immune cells as well as low-expressed hsa-miR-33b-5p were favorable prognostic indicators of overall survival [192]. On the same line of thought, Liu et al. analysed primary tumor, adjacent normal tissues, and matched liver metastases by whole-exome RNA sequencing and SNP6.0 analysis. Their data on gene networks of EMT, angiogenesis, immune-suppression, and T cell exhaustion were closely correlated with the poor patient outcome and intrinsic anti-PD-1 resistance, suggesting a combinational strategy for treatment of metastatic CRC [193].

## 7. EMT, Stemness and CTC

The “embryonal-rest theory” developed by Cohnheim in the 1875 first proposed the notion of cancer stem cells (CSCs), hypothesizing the presence of embryonic-like cancerous cell remnants in adult tissues which then develop into cancer in a non-spontaneous way [194].

The first evidence supporting the idea that EMT markers are also expressed in CSCs dates back to 10 years ago [4,195]. Losing their epithelial characteristics by lack of polarity or intercellular adhesion, CSCs gain mesenchymal properties followed by migration and colonization in different parts of the body [36]. CSCs are implicated in cancer metastasis and chemoresistance [196], and recent data suggest that EMT contributes to cancer metastasis also by increased stemness [197].

Two different models of migrating stem cells were supported by Tsai et al. and Ocana et al. in 2012. The first work supports the concept that upregulation of an EMT activator (e.g., *TWIST1*) in invasive cells of the primary tumor induces dissemination. Its downregulation plus a subsequent re-differentiation (MET) at the distant site would be necessary to allow the colonization and subsequent growth of metastases. Because *TWIST1* also induces stemness properties and growth arrest, putative CSC would be mobile but not proliferating [149]. In the second model, the EMT inducer *PRRX1*, newly identified by Ocana et al., suppresses stemness properties during EMT and dissemination favoring the parallel maintenance of MET, proliferation, and stemness. Accordingly, putative cancer stem cells are not mobile but fixed in the epithelial tumor mass, both in the primary tumor and metastases (“stationary cancer stem cells”). Both models required EMT for dissemination and MET for colonization [198].

Close to local invasion, cells undergoing EMT invade the nearby blood vessels to become circulating tumor cells (CTCs). Since EMT also prevents apoptosis and senescence and contributes to immunosuppression, it is a pleiotropic player along the progression led by the traveler cells [42]. Vitally, cells undergoing EMT can also acquire stem-cell like features. In this scenario, EMT induced cancer stem cells (CSC) contribute to cancer propagation, as shown for pancreatic ductal adenocarcinoma (PDAC) and colorectal cancer (CRC) [61,62,63].

Although every cell could potentially become cancerous, actually only a subset can initiate cancer [199], as CSCs. This is a small subset of cells with intensive proliferative and self-renewal potential, able to differentiate in heterogeneous tumorigenic cells [200]. If able to detach from the primary tumor, access and survive in the circulation, then they can disseminate and transmigrate to form the secondary lesions [201]. Notably, non-CSCs can be converted in CSCs via EMT. This reprogramming is mediated by environmental signals [202,203], derived from stromal cells and forcing cancer cells to undergo EMT and acquire CSCs properties [202]. Furthermore, CTCs can also acquire an aggressive and SC-like phenotype through EMT [204,205]. CSCs markers in the context of CRC include CD133, CD44, CD24, CD166, LGR5, and ALDH-1 [206,207,208,209,210].

Studies on human PDAC also revealed the presence of specific subpopulation with CSCs features. Such cells mainly express CD44, CD24 surface markers, as well as epithelial-specific antigen (ESA) [211,212]. Regarding the self-renewal capability, pancreatic CSCs display the upregulation of important developmental genes that maintain self- renewal in normal stem cells, such as *Sonic hedgehog* (*SHH*) and *BMI-1* [213].

CTCs have been isolated and characterized through liquid biopsy in various tumor types. Zhao et al. demonstrated a positive correlation between bi-phenotypic and mesenchymal, but not epithelial, CTCs and CRC disease stage. These results indicate that CTCs undergoing EMT, or displaying a full mesenchymal phenotype, are associated with more severe disease course. It is thus conceivable that the detection of EMT in CTCs, by panels including epithelial (*EpCAM* and *CK8/18/19*) and mesenchymal (*VIM*, *TWIST1*, *AKT2*, and *SNAI1*) markers may facilitate the recognition of patients at high risk of metastasis [214]. CTCs can accordingly be classified as epithelial (E-CTS), mesenchymal (M-CTS) and epithelial-mesenchymal hybrids (E/M- CTS). The quantification and subtyping of CTCs in the peripheral blood could become a potential noninvasive tool to predict tumor recurrence [215,216].

CTCs in PDAC patients likely exploit the portal circulation to reach the liver, so a comparative analysis of CTCs both in the portal and systemic circulation could help better understand the role of CTCs in metastases and predict prognosis [217]. Tien et al. compared CTC counts between portal vein and peripheral vein blood samples collected during pancreaticoduodenectomy in 41 patients, and reported a higher detection rate in portal than in peripheral blood (58.5% vs. 39%), which was a significant predictor of early development of liver metastases after surgery [218].

Many different methods of CTC isolation have been developed based on their physical properties, such as size exclusion [219] of tumor cells, positive selection by surface markers [220], or negative removal of white blood cells [221]. However, there has not been any unifying method so far [222].

One study employed the pretreatment of blood samples by density gradient centrifugation plus the depletion of CD45+ white blood cells to negatively enrich the recovery of CTCs. This study that aimed to identify distinct EMT states of CTCs in PDAC patients showed that CTC total count and their mesenchymal sub phenotype were high risk factor for the occurrence of postoperative distant metastasis, and early recurrence (within six months) [223].

Moreover, CRC cells that have undergone EMT could be isolated from CTCs and further propagated in culture, and such cells express epithelial and mesenchymal genes as well as stemness markers (i.e., CDH1+, SNAIL+, CD133+) [122].

## 8. EMT and Chemoresistance: An Open Issue

The final step of cancer spreading, if reached, is metastasis development and outgrowth, which will ultimately determine patient fate, and despite the continuous development of surgical and pharmacological treatments, many patients still die from metastasis.

c-MET, a kinase receptor for Hepatocyte growth factor (HGF), drives tumorigenesis repressing the Raf kinase inhibitory protein (RKIP), whose loss has been reported in many cancer types overexpressing c-MET, including 50% of CRC [224]. Several c-MET inhibitors have been developed as anticancer drugs. A phase I study of the c-Met tyrosine kinase inhibitor SAR125844 with MET-amplified solid tumors, demonstrated that the administration of SAR125844, led to a modest antitumor activity. In tumors that had a high level (≥97%) of MET amplification, the objective response rate was 10.5% [225]. Nowadays, more c-MET inhibitors with antitumor activity in GI cancer remain at a pre-clinical stage of development.

The interaction between cancer cells and the surrounding stromal cells in TME play important roles in regulating cancer progression and responsiveness to chemotherapy. Recently, Oshi and coll. investigated the relationship between Annexin A1 (*ANXA1*) mRNA levels and EMT in PDAC, reporting that *ANXA1* expression significantly correlated with the EMT pathway score in both The Cancer Genome Atlas (TCGA) and the GSE57495 cohorts. *ANXA1* high PDACs showed high infiltration of T-helper type 2 cells in the TME, advanced histological grade, high expression of Ki-67 (*MKI67*), and worse prognosis. *ANXA1* expression may be positively correlated with susceptibility to gemcitabine and doxorubicin in primary PDAC cell lines, but negatively with the response to 5-fluorouracil in metastatic PDAC cell lines [226].

To distinguish drug susceptibility in CRC according to TME features, a “Signature associated with FOLFIRI resistance and Microenvironment (SFM)” classification was identified by Zhu et al. In 18 “bulk” and four single-cell RNA-seq datasets, they identified six SFM subtypes (A-F). Among these, SFM-F had higher stromal fraction with EMT phenotype, while SFM-C was characterized as MSI and responsiveness to immunotherapy [227]. Hu et al. reported that CAFs secreted exosomes promote metastasis and chemotherapy resistance by enhancing cell stemness and EMT in CRC [228].

Yang et al. showed in vitro that a TWIST1-induced CSC-like phenotype contributes to develop resistance to irinotecan and enhances migration and invasiveness of colon cancer cells [229]. Skarkova et al. [230] first addressed the expression of EMT markers in CRC cell lines obtained from primary tumors and paired lymph node metastases of three patients. They showed the variation of EMT markers after irinotecan and oxaliplatin treatments, by using qRT-PCR, Western blot, migration assay, and cytotoxicity tests. The results support the increased expression of several markers characteristic of EMT and invasiveness in lymph node metastatic cells, with a significant variability between individual samples. The authors concluded that the combined administration of irinotecan and oxaliplatin decreased cell migration and affected the expression of EMT markers to a varying extent, and that EMT is present in metastatic cells over a phenotypic continuum which is heterogeneously modified upon irinotecan and oxaliplatin treatment.

The treatment and survival prediction of CRC patients could be guided by a panel combining EMT and DNA repair gene expression, which has been developed from the in-silico analysis of the TCGA dataset plus three GEO datasets (GSE39582, GSE17536, and GSE14333). Although with limitations, such as population heterogeneity and the need for validation in large clinical trials, Huang et al. distinguished CRC samples in three clusters by the low or high expression of EMT and DNA repair genes. The best prognosis was identified in Cluster 3, characterized by a low expression of mesenchymal genes (*VIM*, *SNAI1*, *SNAI2*, *TWIST1*, *MMP2*, and *FN1*) and a high expression profile of epithelial and DNA repair genes. Instead, the worst prognosis was associated with the Cluster 2, showing a pattern of high expression in mesenchymal genes and low expression in epithelial and DNA repair genes. However, no significant difference in OS was reported for patients in Cluster 2 and 3, regardless of whether they received adjuvant therapy, while patients in Cluster 1, which had increased expression of epithelial markers but down-regulated DNA repair genes, might benefit from chemotherapy [231].

## 9. Conclusions

The pleiotropic EMT program plays a crucial role in the metastatic cascade, being involved in modulating the plasticity of cancer cells as well as their trafficking and self-renewal. EMT experimental work in animal models has not yet been paralleled by translational applications in human cancers. Besides the inherent complexity of experimentally addressing E⇔M plasticity, a certain degree of skepticism outside the circle of basic scientists is likely to contribute to this gap. However, taking advantage of EMT biomarkers in tissues and in liquid biopsies might yield information on cancer dynamics and progression, allowing prognostic and predictive inferences. Possibly, the integration of morphological parameters with immunological, molecular, and genetic data may provide key elements for stratifying EMT gradients in GI cancers, with the prospect to design more effective therapeutic strategies counteracting the metastatic cascade.

## Figures and Tables

**Figure 1 ijms-22-11469-f001:**
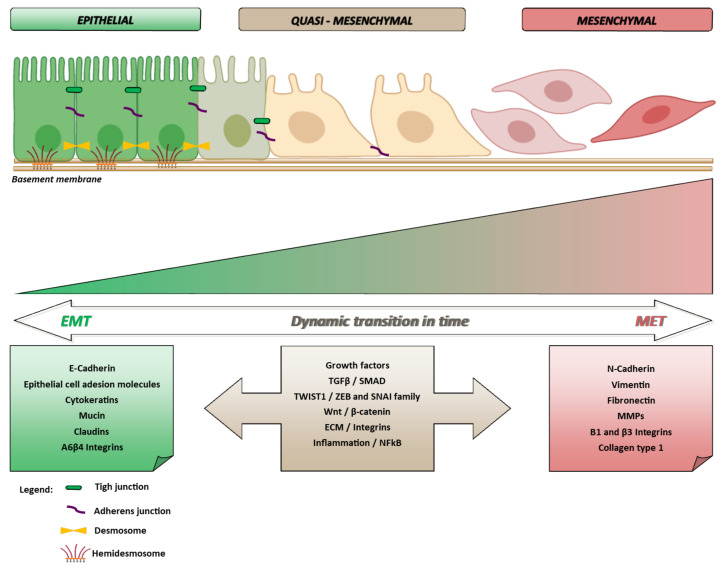
The epithelial phenotype is maintained thanks to the presence of molecules (i.e., CDH1, adhesion molecules, cytokeratins, mucin, integrins and claudins) whose expression allows cells maintaining a base-apical polarization state. The activation of a number of dynamic variables (i.e., nuclear transcription and cell growth factors, signal transduction pathways and activation of inflammatory molecules) can induce the expression of mesenchymal genes. Cells shifting toward a mesenchymal phenotype become mobile and invasive. The transition E⇔M being a reversible process, mesenchymal cells can regain epithelial characteristics. Partially created with BioRender.com (accessed on 14 October 2021).

**Figure 2 ijms-22-11469-f002:**
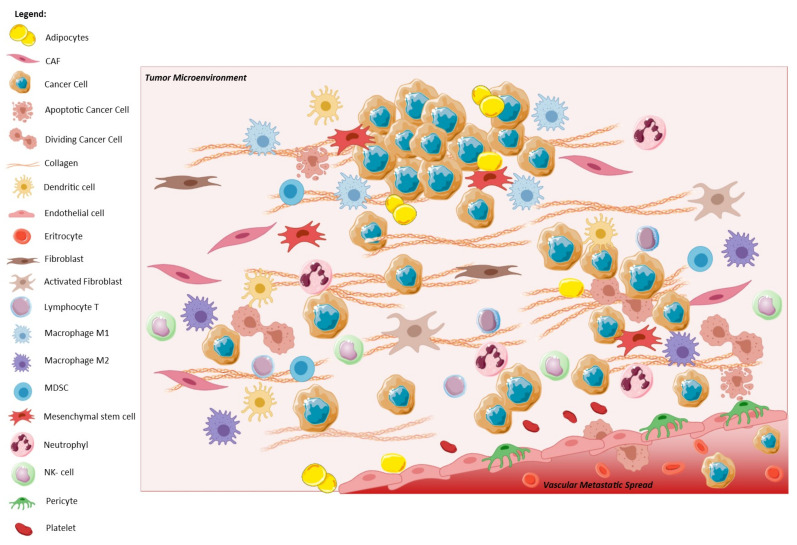
The crosstalk between the tumor and its microenvironment, makes EMT cancer cells able to invade the surrounding stroma, and later the blood vessels, contributing to cancer metastatic spread. The presence of cancer cells leads to the recruitment of many types of cells, like CAFs which activate fibroblasts via TGF-β and secrete extracellular macromolecules, such as collagen. Simultaneously, immune system is activated: dendritic cell, myeloid-derived suppressor cell (MDSC), lymphocyte T and neutrophil, natural-killer (NK) cells, and macrophages (M1 and M2) move against tumor. At the same time, under CAFs signaling, cancer cells undergo to EMT spreading through vessels and contribute to metastatization. Created with Available online: https://smart.servier.com/ (accessed on 14 October 2021).

**Figure 3 ijms-22-11469-f003:**
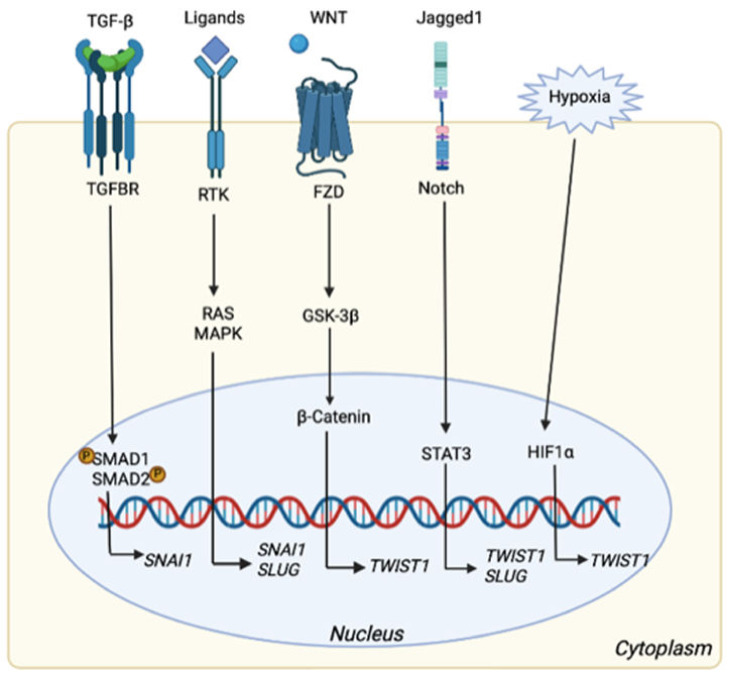
EMT related pathways that can activate the expression of different TFs. Once TGFBR is activated, SMAD1 and 2 are phosphorylated inducing the transcription of *SNAI1* gene. *SNAI1* could also be activated, together with *SLUG* gene, via RAS/MAPK pathways triggered by different ligands (i.e., FGF, EGF). Wnt signaling leads to β-catenin accumulation into the nucleus followed by *TWIST1* transcription, which is also regulated by Notch pathway via STAT3, as well as *SLUG*. Finally, hypoxia contributes to *TWIST1* transcription. Created with Available online: https://biorender.com/ (accessed on 14 October 2021).

**Figure 4 ijms-22-11469-f004:**
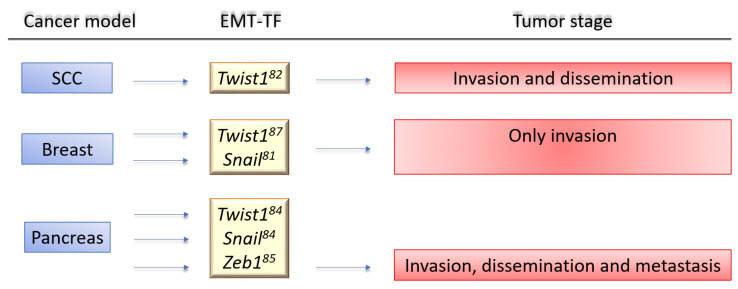
Genetic mouse models to evaluate the relevance of EMT in the metastatic cascade. Three steps (invasion, dissemination, and distant metastasis) are selected. (A) Cancer mouse models based on knock-out (KO) and/or knock-in (KI) of specific EMT-Ts. Ref. [149] *Twist1* conditional KO/KI; Ref. [150] *Snai1* transgene in the context of *H-Ras* activation; Ref. [151] *Snail* conditional KO/KI and *Snail* reporter; Ref. [152] *Snail* or *Twist1* conditional KO; Ref. [153] *Zeb1* conditional KO.

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
