# Peer review of "Epithelial to Mesenchymal Transition: A Challenging Playground for Translational Research. Current Models and Focus on TWIST1 Relevance and Gastrointestinal Cancers"

_ijms, 2021, doi:10.3390/ijms222111469_

Round 1

Reviewer 1 Report

I think this review would have the potential to give a relatively concise overview of the current knowledge in the field concerning EMT, but it may lack some novel elements that connect the rather isolated sections in the current version of the manuscript.  In particular, there is a lack in the "deeper meaning" of EMT and how it is connected by general processes that drive cell fate decisions and (tumor-)cell plasticity. Plus, I am not totally familiar with the specialties of EMT in colorectal cancers, but this may lack deeper, mechanistic insights such as how EMT is related to "lineage-specific differentiation" of human cells and gastrointestinal epithelial cells in particular.

Often, in my opinion, primarily phenotypic changes are described, but the underlying cellular de-differentiation and re-programming mechanisms that are so characteristic for EMT, and drive the plasticity of tumor cells, they are somewhat underrepresented and could go deeper.

This includes even some of the molecular mechanisms that are discussed - but again are relatively well known and represent rather commonplace insights into the genetic "wiring" of EMT: sure, executing transcription factors such as SNAIL; TWIST, and SLUG are involved, but this is now known for decades and per se not very novel. But who regulates SNAIL and SLUG etc.? And there are many cases of "hybrid EMT" in which transcription factors like SNAIL and SLUG are surprisingly NOT involved - but others may that regulate a " propensity of tumor cells to undergo dynamic changes that may often be related to EMT.

NOTCH is mentioned briefly, as a classic cell fate regulator; but that may still represent a relatively late-stage regulatory mechanism of driving any cell types towards mesenchymal functionalities. Nevertheless, NOTCH and WNT are touching the still relatively poorly understood regulation of lineage-specific cell maturation and probably are somewhere not far from the core of "cell fate" decision-making and (tumor)cell plasticity - part of which is leading to EMT. This could be elaborated here. This would have added to the novelty of the article.

For example, sentences like "The EMT process is further regulated by signaling pathways in which TGF-β, Wnt, Notch, and receptor tyrosine kinases (RTKs (co-)operate to modulate the full EMT response" (line 176-177) are in my opinion not very informative, but rather superficial. The question arises: HOW? Arent there any deeper mechanistic insights here? For example, these issues have been investigated already in 2013 in a review article: https://www.nature.com/articles/onc2013128.

There are also other, more recent articles that cover the deeper connection between the inducers of EMT, and their control, such as  PMID: 33808323 and PMID: 30453041. I checked, these and other works are not included in the references. Another very nice paper that does discuss the "deeper connectivity" between EMT and CSCs is PMID: 33968721. In short, I would have expected more discussion on the level of these recent review articles, implementing novel insights from original work. Some of tis may be discussion - not fact; but also not pure speculation. It would also be interesting how gastrointestinal (cancer) cells may engage in this spectrum of phenotypes, and if the corresponding tumors engage in fetal de-differentiation and cell fate decisions. 

Then, some of the following paragraphs on EMT-related genes as biomarkers, actually cover part f this ground - but come across as difficult to read. Much of this is an enumeration of many many different factors, up- or downstream of EMT,  without a clear common "thread" between them that would make this section more readily understandable.  I think the entire section on EMT biomarkers is a bit of a mess and difficult to read. It opens a large number of open issues but it is not trying to connect them functionally, or mechanistically. 

Maybe the best chapters are 7 (on EMT and stem cells) and 8 (EMT and chemoresistance), mainly because they do manage to link clinical and phenotypic issues with molecular and functional associations. Again, some of the deeper connectivity between EMT regulators and regulation of the regulators is outlines here, but a bit disconnected. Because, the focus is again a different one, this time clinical. 

In summary, there are isolated and quite poorly connected chapters or paragraphs in this review article, that all have value by themselves - but some of them seem difficult to read. There needs to be better integration. 

I was hoping the chapters on in vitro modeling may be more informative for the reader. But there are again some very superficial and stylistically not very appealing sections, for example, line 258 - 264. 

The English language use is not always very appealing. There are many expressions (see for examples below) that do not sound fully professional and should be improved - ideally by a native English speaker. Then, some of the sections are not very informative and could be shortened or even omitted. For example, in the introduction, the very 1st paragraph contains almost exclusively commonplaces known to everyone who reads this (= there are changes in gene expression in cancers...) - that would save space if shortened. There are a few other, such places. This also applies, to a lesser degree, to other sections like the description of CSCs (section 2, line 124 - 149). This largely remains somewhat superficial, with a low degree of mechanistic support and little discussion of the clinical consequences or treatment options etc. , which would be generally an area to be considered to bring into the discussion. 

Figure 2: I think Fig.  essentially shows complexity. But it doesnt really help with understanding here the cellular complexity of the tumor tissue (and EMT) really come from and how to bring some order into this mess. Furthermore, the figure legend is full with grammatical errors and should be urgently revised ! Much more than the text, which also requires fixing of grammar and typos alike. 

small things: 

line 42:  the term " losses-and-gain functions" appears very unusual to me. More common would be terms like loss-of-function vs. gain-of-function" or so....

line 69: "The phenotipical effect of EMT".... change to phenotypical

line 106: "could be possible that some cells, considered pro-tumoral, might indeed be cancer cells with EMT features, rather than reactive host cells."  Revise: what is pro-tumoral? 

Author Response

We thank the reviewer. Our rebuttal is attached.

Reviewer 1

Comments and Suggestions for Authors

Q1: I think this review would have the potential to give a relatively concise overview of the current knowledge in the field concerning EMT, but it may lack some novel elements that connect the rather isolated sections in the current version of the manuscript.  In particular, there is a lack in the "deeper meaning" of EMT and how it is connected by general processes that drive cell fate decisions and (tumor-)cell plasticity. Plus, I am not totally familiar with the specialties of EMT in colorectal cancers, but this may lack deeper, mechanistic insights such as how EMT is related to "lineage-specific differentiation" of human cells and gastrointestinal epithelial cells in particular.

A1: We thank the reviewer for this suggestion. We deepened the “lineage-specific differentiation”, in cell maturation and at gastrointestinal tract. We elaborated this topic in the new paragraph entitled “4.1 EMT is related to "The lineage-specific differentiation in gastrointestinal tract”." 

Q2: Often, in my opinion, primarily phenotypic changes are described, but the underlying cellular de-differentiation and re-programming mechanisms that are so characteristic for EMT, and drive the plasticity of tumor cells, they are somewhat underrepresented and could go deeper. This includes even some of the molecular mechanisms that are discussed - but again are relatively well known and represent rather commonplace insights into the genetic "wiring" of EMT: sure, executing transcription factors such as SNAIL; TWIST, and SLUG are involved, but this is now known for decades and per se not very novel. But who regulates SNAIL and SLUG etc.?

A2: We thank the reviewer to arise this point giving us the opportunity to better explain the purpose of our review. The main goal of this article is to highlight the translational aspect of EMT in gastrointestinal cancers, for this reason we only cited some important mechanisms. However, the different regulation of transcription factors is reported in different section, depending on the topic proposed.

Regarding your question about factors regulating SNAIL and SLUG, it is reported the MAPK signaling cascade from the new lines 290-294, then in the following paragraph about miRNA activities (new line 372 and subsequent)

Q3. And there are many cases of "hybrid EMT" in which transcription factors like SNAIL and SLUG are surprisingly NOT involved - but others may that regulate a " propensity of tumor cells to undergo dynamic changes that may often be related to EMT.

A3. We agreed with the reviewer that SNAI1 and 2 are not always involved, in fact the expression of TFs seems to be tumor specific. As previously mentioned, you can find the information related “Hybrid EMT” in chapter 8. Following the reviewer suggestion, we add some additional information: “Zhao et al. demonstrated a positively correlation between bi-phenotypic and mesenchymal, but not epithelial, CTCs, and CRC disease stage. These results indicated that CTCs undergoing EMT, or displaying a full mesenchymal phenotype, suggesting more severe disease. It is thus conceivable that detection of EMT markers in CTCs, as well as of epithelial ones (EpCAM and CK8/18/19) and mesenchymal (VIM, TWIST1, AKT2 and SNAI1) may facilitate the recognition of patients at higher risk of metastasis (Ref 28030836)”. This is reported in the new main text.

Q4. NOTCH is mentioned briefly, as a classic cell fate regulator; but that may still represent a relatively late-stage regulatory mechanism of driving any cell types towards mesenchymal functionalities. Nevertheless, NOTCH and WNT are touching the still relatively poorly understood regulation of lineage-specific cell maturation and probably are somewhere not far from the core of "cell fate" decision-making and (tumor)cell plasticity - part of which is leading to EMT. This could be elaborated here. This would have added to the novelty of the article.

For example, sentences like "The EMT process is further regulated by signaling pathways in which TGF-β, Wnt, Notch, and receptor tyrosine kinases (RTKs (co-)operate to modulate the full EMT response" (line 176-177) are in my opinion not very informative, but rather superficial. The question arises: HOW? Arent there any deeper mechanistic insights here? For example, these issues have been investigated already in 2013 in a review article: https://www.nature.com/articles/onc2013128.

A4. We thank the reviewer for this suggestion. We deepened the Notch and Wnt pathways, focusing on their involvement in the regulation of lineage-specific cell maturation in the gastrointestinal tract. We elaborated this topic in the new paragraph entitled “4.1 EMT is related to "The lineage-specific differentiation in gastrointestinal tract." 

Q5. There are also other, more recent articles that cover the deeper connection between the inducers of EMT, and their control, such as  PMID: 33808323 and PMID: 30453041.

A5. We thank for the suggestion, but we prefer to report the original work that covers the deeper connection between the inducers of EMT, alike our new refs 173-174.

Q6. I checked, these and other works are not included in the references. Another very nice paper that does discuss the "deeper connectivity" between EMT and CSCs is PMID: 33968721. In short, I would have expected more discussion on the level of these recent review articles, implementing novel insights from original work. Some of tis may be discussion - not fact; but also not pure speculation. It would also be interesting how gastrointestinal (cancer) cells may engage in this spectrum of phenotypes, and if the corresponding tumors engage in fetal de-differentiation and cell fate decisions. 

Then, some of the following paragraphs on EMT-related genes as biomarkers, actually cover part f this ground - but come across as difficult to read. Much of this is an enumeration of many many different factors, up- or downstream of EMT,  without a clear common "thread" between them that would make this section more readily understandable.  I think the entire section on EMT biomarkers is a bit of a mess and difficult to read. It opens a large number of open issues but it is not trying to connect them functionally, or mechanistically. 

A6. It is understood the criticism raised by the Reviewer. However, our review tries to push on the relevance of the function of EMT in the metastatic cascade, rather than on the mechanisms. Clearly, this might not always be satisfying, but we believe that EMT has been disregarded in translational sciences, and it needs more attention, as a relevant process, although mechanisms remain to be explained. As to metastasis, there is a need for new parameters to be investigated in translational science beside genetic and epigenetic damage, and EMT is one the main candidates. Accordingly, we deepened the coverage of the mechanisms, but did not change our focus on EMT as a leading candidate requiring more support than explanations.

Maybe the best chapters are 7 (on EMT and stem cells) and 8 (EMT and chemoresistance), mainly because they do manage to link clinical and phenotypic issues with molecular and functional associations. Again, some of the deeper connectivity between EMT regulators and regulation of the regulators is outlines here, but a bit disconnected. Because, the focus is again a different one, this time clinical. 

In summary, there are isolated and quite poorly connected chapters or paragraphs in this review article, that all have value by themselves - but some of them seem difficult to read. There needs to be better integration. 

Q7. I was hoping the chapters on in vitro modeling may be more informative for the reader. But there are again some very superficial and stylistically not very appealing sections, for example, line 258 - 264. 

A7. We thank the reviewer and we made more informative the chapters on in vitro modeling.

Q8. The English language use is not always very appealing. There are many expressions (see for examples below) that do not sound fully professional and should be improved - ideally by a native English speaker. Then, some of the sections are not very informative and could be shortened or even omitted. For example, in the introduction, the very 1st paragraph contains almost exclusively commonplaces known to everyone who reads this (= there are changes in gene expression in cancers...) - that would save space if shortened. There are a few other, such places. This also applies, to a lesser degree, to other sections like the description of CSCs (section 2, line 124 - 149). This largely remains somewhat superficial, with a low degree of mechanistic support and little discussion of the clinical consequences or treatment options etc. , which would be generally an area to be considered to bring into the discussion. 

A8. We thank the reviewer and we made more informative the chapters on CSCs

Q9. Figure 2: I think Fig.  essentially shows complexity. But it doesnt really help with understanding here the cellular complexity of the tumor tissue (and EMT) really come from and how to bring some order into this mess. Furthermore, the figure legend is full with grammatical errors and should be urgently revised ! Much more than the text, which also requires fixing of grammar and typos alike. 

small things: 

line 42:  the term " losses-and-gain functions" appears very unusual to me. More common would be terms like loss-of-function vs. gain-of-function" or so....

line 69: "The phenotipical effect of EMT".... change to phenotypical

line 106: "could be possible that some cells, considered pro-tumoral, might indeed be cancer cells with EMT features, rather than reactive host cells."  Revise: what is pro-tumoral? 

A9. For all small things and figure 2: we have corrected grammar and typos errors and modified legend and caption of figure 2

By referring to “some cells, considered pro-tumoral, might indeed be cancer cells” we meant cancer cells in mesenchymal disguise, alike CAFs (as shown by Celesti G, Gastroenterology, 2013).

We finally thank the reviewer for his contribution in the revision.

Reviewer 2 Report

Dear Dr.

Editor,

Overall recommendation:

 Accept

Final comments:

  This paper shows how epithelial mesenchymal transition (EMT) occurs in gastrointestinal cancers. Their data looks very clear and interesting. These data prompt us that prevention of EMT shold be a good target for treating cancers. If the authors have any idea how to inhibit EMT please add discussions.

I think this paper is good for publication in this present form, discussions about clinical applications will attract gastro-enterologists.

Kansai Medical University

Katsunori Yoshida

Author Response

We thank the reviewer. Our rebuttal is attached.

Comments and Suggestions for Authors

Dear Dr.

Editor,

Overall recommendation:

 Accept

Final comments:

Q1. This paper shows how epithelial mesenchymal transition (EMT) occurs in gastrointestinal cancers. Their data looks very clear and interesting. These data prompt us that prevention of EMT shold be a good target for treating cancers. If the authors have any idea how to inhibit EMT please add discussions.

A1. We thank the reviewer for his opinion. In consideration of his request and the one submitted by reviewer 3, we have developed an introductory chapter in which we have also reported a possible answer to your question.

I think this paper is good for publication in this present form, discussions about clinical applications will attract gastro-enterologists.

We thank the reviewer for his contribution in the revision.

Kansai Medical University

Katsunori Yoshida

Reviewer 3 Report

Review

Epithelial to mesenchymal transition: a challenging playground for translational research. Focus on gastro-intestinal cancers.

Luana Greco, Federica Rubbino, Alessandra Morelli, Federica Gaiani, Fabio Grizzi, Gian Luigi de’Angelis, Alberto Malesci and Luigi Laghi

Journal: International Journal of Molecular Sciences

In this review, the authors discuss about the recent knowledge on EMT, a critical cell biological process that triggers deep morphological, molecular and biochemical changes, important because it is nowadays well established that most tumour cells highjack the EMT programme to progress towards malignancy. The authors focus on molecular mechanisms which become activated during EMT. The authors report that the focus of the review is on gastro-intestinal cancers.

The paper results well-structured and it fits well in the panorama of the flourishing literature on the subject.

The only objection I would like to raise is that the manuscript does not appear to be clearly focused on gastro-intestinal cancers. Although the authors frequently refer to this type of cancers, an introductory paragraph is missing that could, for example, summarize the most recent findings on the role of EMT in this specific, albeit very diverse, class of cancers. I leave to the authors to choose how to structure this paragraph and its place in the review.

Author Response

We thank the reviewer. Our rebuttal is attached.

Journal: International Journal of Molecular Sciences

In this review, the authors discuss about the recent knowledge on EMT, a critical cell biological process that triggers deep morphological, molecular and biochemical changes, important because it is nowadays well established that most tumour cells highjack the EMT programme to progress towards malignancy. The authors focus on molecular mechanisms which become activated during EMT. The authors report that the focus of the review is on gastro-intestinal cancers.

The paper results well-structured and it fits well in the panorama of the flourishing literature on the subject.

Q.1 The only objection I would like to raise is that the manuscript does not appear to be clearly focused on gastro-intestinal cancers. Although the authors frequently refer to this type of cancers, an introductory paragraph is missing that could, for example, summarize the most recent findings on the role of EMT in this specific, albeit very diverse, class of cancers. I leave to the authors to choose how to structure this paragraph and its place in the review.

A1. We thank the reviewer to arise this point giving us the opportunity to better explain the role of EMT in this specific, albeit very diverse, class of cancers and the state of the art of knowledge is critical for them. Following this suggestion, we have developed an introductory paragraph on the state of the art of knowledge to date on EMT, CRC and PDAC. This paragraph contains the purpose of the review, summarizing what is known at a translational level in relation to EMT and the development of the two neoplasia, to suggest points of discussion for clinical applications.

We thank the reviewer for his contribution in the revision.

Round 2

Reviewer 1 Report

This article gets now a bit out of hand - and especially, it gets far too long. And it is beginning to lose focus, especially now that so many new sections have been added - in response to some of the reviewer comments. In my opinion, this is one of the rare cases where a peer-review process does NOT necessarily lead to an improvement of the article - but even further complicates the issue by leading to a loss of orientation and focus. 

Without going into every detail that is to criticize there, I would rather summarize the main issues: 

the NEW introduction is somewhat odd, it is unusual for an introduction to describe details about certain EMT-related proteins such as fibronectin, rather than illustrating the "history" of EMT research in colorectal cancer CRC. 

The "old" introduction is still doing that job, and I don't understand what is the thinking of the authors, why they have now composed a new intro section that isn't really an introduction at all - but rather a facet of EMT that cold easily come later. 

There is a new chapter on EMT in connection to lineage-specific differentiation (introduced after my suggestion) which I appreciate. But what becomes now quite apparent is that there is no logical order in the chapters: the role of EMT in stem cell biology comes BEFORE the lineage-specific differentiation, I would bring that quite late. Maybe after a longer description of the phenotypic consequences of EMT for tumor cells: this is spread over different chapters, it could actually justify for a chapter on its own - because this would summarize why EMT is such an important process: it makes tumor cells more aggressive, more invasive, more motile, and contributes to the tumor-cell plasticity that is such a critical issue for things like drug resistance, tumor dormancy, invasion and metastasis, maybe vascular mimicry, intravasation, and extravasation, and last not least, of course, stemness. I am kind of missing that logical order of how EMT and its consequences are described. 

Instead, there are rather isolated chapters on various issues, that loosely relate to each other, but there is no clear narrative and a logic flow. 

What is maybe most critical: the review gets way too long (almost 30 pages).  Especially now, that there have been several lengthy additions, the page limit should be reached - actually inviting the review to be split into at least 2 independent articles. That would also provide much more focus and give a clear logical narrative. For example, various models of EMT from spheroids and organoids to animal models could clearly provide one story on its own. Then, there is a long discussion about EMT-related genes and biomarkers, including transcription factors, expressed in tumor tissues. This chapter could easily be somewhat expanded, as there is a large number of articles that correlate certain EMT-related biomarkers with phenotypic or histologic features in tumor tissues, including phenomena at the invasive front, and issues such as "tumor budding" etc. that are currently discussed widely in the field. This would be more than sufficient material for a review on its own. 

Furthermore, focus is lost by discussing some EMT-related transcription factors such as TWIST in almost every possible detail.... over 2 pages of it have been added. But this raises the question: why not discuss other factors such as snail and slug (SNAI1 and 2) or ZEB1 in similar detail? And maybe add something of their consequences, such as which are the known target genes, etc. At least, it should be disclosed and justified why TWIST1 and 2 get so much attention, why they may be more critical in CRC than maybe in other tumor types? I dont know, I am looking for this info and its not there. Instead, there is a lot of detailed and (in principle) well-researched material here, that could be even further expanded, maybe cleaned up a little, and put into more logical order - and you would have another, independent review article there. 

There is also a sufficient bulk of information concerning model systems to investigate EMT - covering both in vitro (organoids, spheroids) and in vivo models animal models, xenografts, genetically engineered mouse models etc). This section is by itself large and informative enough to be considered as a spin-off article on its own. It could be expanded - especially the issue with patient-derived organoids versus cell-line based organoids needs to be covered in more detail. But this has potential as an article on its own. 

The story concerning the many different models is also significantly different from the histopathological issues, in fact there are not many links between them, so it is unclear to me why the HAVE to be joined in the same, lengthy article. I would rather considering breaking the text apart. 

In short - my major concern with this review is that it is getting waaaaay too long, and also increasingly loses focus. That's of course a question to be addressed by the editors. But my suggestion would be to split it and make at least 2 reviews out of this massive piece of work that is difficult to read. Why not use the current body of work and get 2 articles instead of one? 

There are many other issues, related to wording and grammar, that also still need the attention of a native English speaker, or someone editing the text. Especially in some of the newer passages, there are now many of these grammar problems, typos, stylistic issues, too many to an outline without actually becoming a co-author of the manuscript. 

Author Response

Our rebuttal is attached 

Round 3

Reviewer 1 Report

The authors have now significantly streamlined and somewhat re-arranged the manuscript, and also put some of the dominant elements (such as the focus on TWIST) into more context, so it makes a lot more sense. They have also changed, or clarified the title, which is a good thing. 

In my opinion, it is still too long for a "standard" review article, but this is the matter of the editors to decide. It has now also clearly become more INFORMATIVE, even if it is a big chunk of information, but this is what a review article is supposed to convey. Therefore, I think it should now soon be ready for publication, after fixing a few, final issues here and there - most of them language-related: There are STILL some English-language issues, but this time it appears more likely they can be fixed in production.